# Causality is Invariance Across Heterogeneous Units

## Abstract

Learning a model from data for the three layers of Pearl Causal Hierarchy (PCH) (i.e., the associational, the interventional, and the counterfactual) is a central task in contemporary causal inference research, and it becomes particularly challenging for counterfactual queries. The prevailing scientific understanding is anchored in the three-step counterfactual algorithm (i.e., abduction, action, and prediction) proposed by Judea Pearl, which he considers as one of his most pivotal contributions. While this algorithm offers a theoretical solution, the absence of complete knowledge on structural causal models (SCMs) renders it highly impractical in most scenarios. To tackle the tasks of PCH, this paper introduces the DiscoModel, grounded in the core principle that "Causality is invariance across heterogeneous units." The underlying causal modeling theory of our model is *Distribution-consistency Structural Causal Models* (DiscoSCMs), which extends both *structural causal models* and the potential outcome framework. The model is implemented through a customized neural network, comprising two sub-networks: AbductionNet and ActionNet. The former infers the selection variable on heterogeneous units, while the latter encapsulates the invariant causal relationship. DiscoModel exhibits remarkable capability for all the three layers of PCH simultaneously, providing practical and reasonable answers to important counterfactual questions (e.g., "For a user on a specific internet platform who has been observed with high subsidy and high retention, would returning to the past and continuing to provide a high subsidy result in high retention?"). To the best of our knowledge, DiscoModel is the first to provide non-trivial answers to such queries, substantiated through experiments on both simulated and real-world data.

## 1 Introduction

In the field of causal modeling, there are two primary frameworks: Potential Outcomes (PO) (Rubin, 1974; Holland, 1986) and Structural Causal Models (SCMs) (Pearl, 1995; 2009). The former represents "experimental" causality for individuals, while the latter casts causal inference as Pearl Causal Hierarchy (PCH) (Pearl & Mackenzie, 2018; Bareinboim et al., 2022) consisting of three layers: the associational (Layer 1), the interventional (Layer 2), and the counterfactual (Layer 3). However, these two frameworks, which are mathematically equivalent, encounter the issue of degenerate counterfactuals (Gong, 2023). This problem arises from *the consistency assumption*, i.e., the potential outcome $Y(t)$ must be consistent with y when observing $T = t$ and $Y = y$. This issue makes it very challenging to establish a practical model for Layer 3 Valuations under the current mainstream causal modeling frameworks, as it requires complete knowledge of structural equations among domain variables.

Consider a scenario where a user on a specific internet platform is observed with high subsidy and high retention. A pertinent question arises: "*Would the user still have demonstrated high engagement if we go back in time and still provide a high subsidy, assuming all other conditions remained equal?*" The observed high retention could be attributed either to the generous subsidy or simply to fortuitous circumstances. Regardless, models developed under conventional frameworks would consistently predict high retention due to the

consistency assumption, thereby rendering the model impractical when acknowledging that good fortune is something that we cannot control. In contrast, employing DiscoSCM allows for the prediction of low retention, with the probability of such an outcome varying across heterogeneous units (Gong, 2023).

In this paper, we introduce a practical model learned from data, DiscoModel, which is capable of conducting Layer 1/2/3 Valuations simultaneously. The theoretical framework of the model we use is DiscoSCM, serving as an extension of both aforementioned frameworks. The fundamental idea behind it is "Causality is Invariance Across Heterogeneous Units", and the specific algorithm employed is the Population-level Counterfactual Algorithm. In terms of implementation, the system consists of two specialized sub-networks: AbductionNet and ActionNet. AbductionNet is used for inferring the unit selection variable $S$, while ActionNet is for computing the unit potential outcome. Finally, it has been validated that DiscoModel is a reasonable and practical model for Layer Valuations through experiments on both simulated and real data.

## 1.1 Related Work

Pearl (Pearl, 1995; 2009; Pearl & Mackenzie, 2018; Pearl, 2018) proposes a division of causal information into three distinct layers, forming a three-layer causal hierarchy. These layers are 1. Association, 2. Intervention, and 3. Counterfactual, each corresponding to their respective usage. Questions at Layer $i$ (where $i = 1, 2, 3$) can only be answered if information from Layer $i$ or higher is available, which recently has been rigorously defined mathematically as Layer Valuations (Bareinboim et al., 2022). Generally, Layer Valuations require complete knowledge of structural equations in the underlying SCM, making it impractical for most real-world applications. Efforts have been made to learn these valuations from data using neural nets (Xia et al., 2021). However, learning the complete causal graph among domain variables and deterministic structural equations from data remains a significant challenge. Furthermore, Even with complete knowledge of SCM, there are no straightforward methods for practically calculating Layer 3 Valuations with the three-step counterfactual algorithm proposed by Pearl (2009). Addressing these challenges appears to be fraught with considerable difficulties.

Gong (2023) realizes that the issue of individual-level degenerate counterfactuals is essentially caused by the consistency assumption, and thus addresses it theoretically by extending the SCM framework to DiscoSCM. They prove that in DiscoSCMs, under the condition of independent potential noise, individual-level counterfactuals at Layer 3 can be directly reduced to Layer 2 valuation calculations. Besides, they further theoretically prove that population-level counterfactuals can be computed through a three-step algorithm: abduction, action, and reduction. Due to space constraints, further content related to this work are relegated to the Appendix.

## 2 DiscoModel

In this section, we introduce DiscoModel, a model grounded in the core principle that *causality is invariance across heterogeneous units.* This principle is crucial, especially in fields like social science where population heterogeneity is ubiquitous. It implies that while different units share a common mechanism for generating unobserved outcome parameters, variations in parameters are attributed to each unit's unique causal representation. It is implemented through a customized neural network, consisting of two sub-networks: AbductionNet and ActionNet. AbductionNet infers the selection variable on heterogeneous units, while ActionNet encapsulates the invariant causal relationship. Initially, we will clarify the settings, assumptions, and the associated Layer Valuation algorithm framework underlying the DiscoModel.

## 2.1 Settings, Assumptions and Mathematical Foundations for DiscoModel

Consider a common causal modeling setting involving domain variables: treatment $T$, pre-treatment features $X$, and outcomes $Y$. To perform counterfactual predictions, i.e., $P(Y(t) = y|e)$, Pearl (2009) proposes a three-step process:

1. Abduction: Update the probability $P(u)$ to obtain $P(u|e)$.
2. Action: Modify the equations determining the variable $T$ to $T = t$.
3. Prediction: Utilize the modified model to compute the probability $P(Y = y)$.

Here, $e$ represents the observed trace, also referred to as evidence, exemplified by $X = \tilde{x}, T = \tilde{t}, Y = \tilde{y}$, which can be succinctly represented as $e = [\tilde{x}, \tilde{t}, \tilde{y}]$. When there is no observed trace for any domain variables, this is denoted as $e = []$. The corresponding random variable for the counterfactual distribution $P(Y(t) = y|e)$ is called the counterfactual variable, denoted as $Y(t|e)$, representing the outcome under intervention $t$ given the evidence $e$. For conciseness, the use of bold symbols is avoided unless there is ambiguity or a specific need to emphasize that they are vectors.

As stated in Pearl (2018) (p. 3), the vector $U = u$ can be interpreted as an experimental "unit" which can stand for an individual subject, and every instantiation $U = u$ of the exogenous variables uniquely specifies all domain variable values. While Pearl's counterfactual algorithm requires complete knowledge of structural equations, limiting its practical application, DiscoSCM innovatively decouples individual semantics from the exogenous variable $U = u$, assuming the same distribution of remaining exogenous uncertainty across all individuals. Such separation calls for distinct algorithms for individual-level and population-level calculations. Specifically, for individual-level valuations under DiscoSCM with independent potential noises(Gong, 2023):

$$P(Y_i(t|e) = y) \triangleq P(Y_i(t) = y|e) = P(Y_i(t) = y) \tag{1}$$

for any individual $i$ in a population, such as all users of an online platform. Let $S$ be the unit selection variable for this population, which is typically assigned a uniform prior distribution. Consequently, for population-level calculations, a corresponding three-step algorithm is employed:

---

**Algorithm 1:** Population-Level Counterfactual Algorithm

---

1: **Abduction:** Compute the posterior $P(\cdot|e)$ of the unit selection variable $S$, denoted its corresponding variable as $S(e)$.
2: **Valuation:** Derive the individual-level counterfactual variable $Y_i(t|e)$ with certain method for each unit $i$.
3: **Reduction:** Obtain an estimation of the sub-population counterfactual outcome

$$Y(t|e) = Y_{S(e)}(t) \tag{2}$$

using a reduction method such as expectation or direct sampling.

---

The formula in the reduction step of Algorithm 1 for population-level Layer Valuations can be justified as follows:

$$
\begin{align}
P(Y(t|e) = y) &= P(Y(t) = y|e) && \text{by definition} \tag{3}\\
&= \sum_i P(Y_i(t) = y|e, i)P(i|e) && \text{by probability formula} \tag{4}\\
&= \sum_i P(Y_i(t) = y)P(i|e) && \text{by equation 1} \tag{5}\\
&= P(Y_{S(e)}(t) = y) && \text{by probability formula} \tag{6}
\end{align}
$$

This innovative algorithm for Layer valuations inspired the development of DiscoModel, which consists of two sub-networks: AbductionNet and ActionNet. AbductionNet is used to

infer the probability of a user being identified when $X = x, T = t, Y = y$ is observed, namely, $P(i|e) = P(S = i|X = x, T = t, Y = y) \triangleq P(S(e) = i)$. Meanwhile, ActionNet directly computes the causal parameters for any unit $i$ with its causal representation $z_i$. Specifically, AbductionNet, with $e$ as the input and posterior $P(\cdot|e)$ as the output, addresses the question: "Given the evidence $e$, how do we calculate the probability of selecting a particular unit $i$?" To do this, it employs a highly sophisticated design based on the following assumption:

**Assumption 1** (Unit Causal Representation). *For each individual $i$, there exists a causal representation $z_i$ satisfying that,*

$$P(Y(t) = y|e, S(e) = i) = P(Y(t) = y|e, z_{S(e)} = z_i) \tag{7}$$

*for any treatment $t$, outcome $y$ and evidence $e$.*

This assumption suggest that, for individual $i$, the individual-level counterfactual outcome shares the same distribution as the counterfactual outcome for the sub-population possessing a similar causal representation. Hence, to calculate counterfactuals, we only need a model that predicts the outcome from the causal representation and computes the posterior for causal representation. Both of these can be learned with population data, thereby underscoring the pivotal role of causal representation to overcome the computational difficulties on estimating counterfactuals.

For simplicity, we may assume that the unit representation variable follows a prior of a standard multivariate normal distribution, namely $z_S \sim N(0, I)$. The rationale of the network's structural design can be revealed itself through extreme cases. Firstly, in the extreme case where no observational information is present, i.e., $e = []$, the posterior should naturally align with the prior. This implies that drawing a sample $z$ from a standard multivariate normal distribution is equivalent to randomly selecting a sample $i$ from the population. Secondly, in the opposite extreme where $e$ contains abundant information, enough to determine which unit $i$ has been selected, $S(e)$ should converge to a Dirac distribution at $i$. Consequently, $z_{S(e)}$ should be approximated by a distribution with mean $z_i$ and variance near zero. These scenarios guide the setting of the output of AbductionNet as a multivariate normal distribution, serving as the distribution for the causal representation variable $z_{S(e)}$.

More specifically, the AbductionNet is designed as a mapping $f : e \to \boldsymbol{\mu}_r, \boldsymbol{\Sigma}_r$, serving as the parameters for the causal representation variable $z_{S(e)}$ with multivariate normal distribution. Meanwhile, ActionNet is also a mapping $g : z, t \to \boldsymbol{\mu}_o, \boldsymbol{\Sigma}_o$, serving as the distribution parameters for the unit potential outcome variable $Y_i(t)$ with normal distribution. Essentially, DiscoModel aims to learn the functions $f$ and $g$ from the data, ultimately for the estimation of counterfactual outcome $Y(t|e)$. To gain a clearer understanding of the design of DiscoModel, we will now delve into the details of the network architecture in the following subsection.

## 2.2 Network Structure and Components

Our DiscoModel, as illustrated in Fig. 1, primarily consists of two sub-networks: AbductionNet and ActionNet. Delving into the architecture, AbductionNet is composed of several layers. It features a sequence of TYGatedInjectBlocks (refer to Fig. 2b) and an MLP layer, succeeded by a NormalVectorLayer. This last layer models the distribution of the unit causal representation variable for sub-population $S(e)$, which is assumed to follow a normal distribution $\mathcal{N}(\boldsymbol{\mu}_r(e), \boldsymbol{\Sigma}_r(e))$. The network processes the observation $e$ to infer user representations, which subsequently serve as input to ActionNet. On the other hand, ActionNet employs the unit causal representations inferred by the AbductionNet to calculate the parameters of the outcome variables. This network is comprised of a sequence of GatedInjectBlocks (See Fig. 2a) and an MLP layer, culminating in a NormalVectorLayer. The latter is used to ascertain the distribution of $Y_i(t)$, presumed to be $\mathcal{N}(\boldsymbol{\mu}_o(z_i, t), \boldsymbol{\Sigma}_o(z_i, t))$.

A distinctive feature of DiscoModel is its ability to takes a triplet $[x, t, y]$ or its subset as input, in contrast to other causal representation learning methods, such as CEVAE (Louizos et al., 2017), TARNet (Shalit et al., 2017), and DragonNet(Shi et al., 2019), which typically not take $y$ as input. Consequently, a natural question arises: "Why do we opt to use $[x, t, y]$ as input to infer causal representation? Additionally, how is this feature achieved?" To

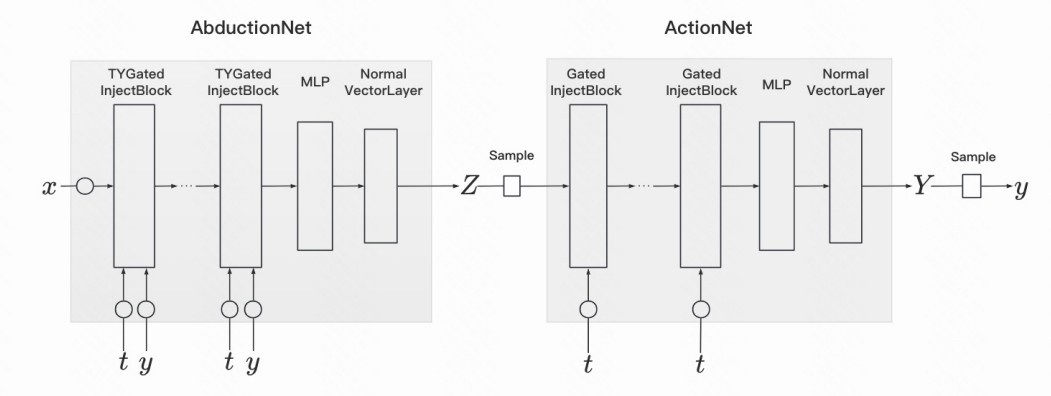

Figure 1: The DiscoModel, consisting of two sub-networks: AbductionNet and ActionNet. The former is responsible for deriving ausal representation variable $z_{S(e)}$, while the latter computes the unit potential outcome variable $Y_i(t)$.

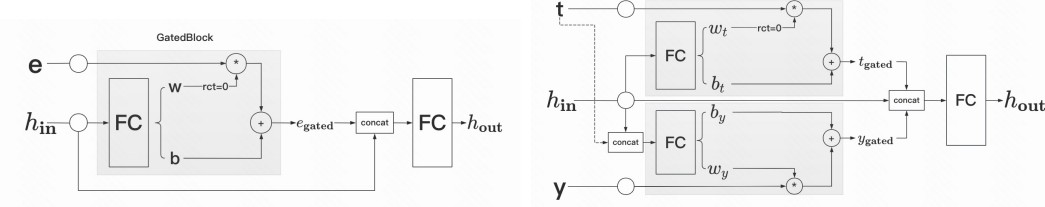

(a) The GatedInjectBlocks, featuring a gate for observation $e$, is a fundamental module capable of controlling the input of specific information.

(b) The TYGatedInjectBlock consists of two gates for $t$ and $y$, enabling flexible control of the input information of observed $t, y$. It is a core component of AbductionNet.

Figure 2: Building blocks for DiscoModel.

illustrate, we may consider unobserved cancer genes as the causal representation $z$, physical examination results as $x$, and treatment $T$ and outcome $Y$ as smoking status and cancer diagnosis, respectively, then under the condition that it is known that the individual smokes and has been diagnosed with cancer, it is evident that incorporating $t, y$ information can more accurately infer $z$. Therefore, the input for AbductionNet is setting to $[x, t, y]$ rather than just $x$. The key to our ability to achieve this lies in a Gated Design in GatedInject-Blocks/TYGatedInjectBlocks (See Fig. 2). In a nutshell, this is accomplished by filling None in $[x, t, y]$ with thoughtfully crafted bias terms, represented by empty circles presented in Fig. 1 and Fig. 2. For more detailed information, please refer to the corresponding code implementation.

Another pivotal structural design incorporated is the *Multiple Inject Layers*, a structure devised to prevent the signal of $t$ from being overwhelmed. This design ensures that the network's forward propagation layers, transitioning from high-dimensional input features $x$ to the output parameters, outnumber those from $t, y$ to the output parameters. This design mitigates the vanishing gradient problem by strategically reducing the forward propagation layers from $t, y$ to the output. Furthermore, it employs multiple gates to fine-tune the impact of $t, y$ on the outcome parameters, thereby enhancing the model's proficiency in processing information pertinent to $t$ and $y$. Additionally, the *NormalVectorLayer* is instrumental for modeling the distribution of the output, assuming a Normal distribution. Initialized with input and output dimensions, it outputs the parameters of the Normal distribution, capturing the inherent variability in the data and facilitating robust predictions.

While some may posit that the application of deep networks for causal inference, exemplified by DiscoModel, is no longer a novelty (Li et al., 2023), we wish to underscore a key differ-

entiation. The prevalent networks are predominantly employed for causal effect estimation, and fall short of addressing Layer 3 counterfactual estimation. Distinctively, DiscoModel exhibits the capability to concurrently conduct Layer 1/2/3 valuations at both individual and population levels. This encompasses counterfactual outcome prediction, potential outcome prediction, individual treatment effect (ITE) estimation, and association prediction, all of which will be further delineated in the ensuing section.

## 3 LAYER VALUATIONS

Our DiscoModel clearly reflects the distinction between individual-level and population-level valuations. The AbductionNet is designed to infer representations of sub-populations, emphasizing the idea that abduction inference is inherently tied to populations. In contrast, the ActionNet is dedicated to computing outcomes for individual, highlighting the principle that actions or interventions are fundamentally rooted at the individual level. Hence, once we have a well-trained model, the individual-level counterfactual $Y_i(t|e) = Y_i(t)$ can be naively computed by any estimation of the potential outcome. On the other hand, estimating the population-level counterfactual involves a three-stage process of abduction, action and reduction. Specifically, for evidence $e = [\tilde{x}, \tilde{t}, \tilde{y}]$ or any subset of it, the counterfactual estimation procedure is formally described as Algorithm 2.

---

**Algorithm 2:** Population-Level Counterfactual Estimation using DiscoModel

---

1: **Abduction:** Use the AbductionNet to compute the distribution parameters of the representation for sub-population $S(e)$, i.e., $\boldsymbol{\mu}_r(e)$ and $\boldsymbol{\Sigma}_r(e)$.
2: **Action:** Use the ActionNet to compute the distribution parameters for the counterfactual outcome $Y_i(t|e) = Y_i(t)$, i.e., $\boldsymbol{\mu}_o(z, t)$ and $\boldsymbol{\Sigma}_o(z, t)$.
3: **Reduction:** Apply a reduction method, such as expectation, mode, or a simple sampling, to derive estimation $\widehat{Y(t|e)}$ for counterfactual outcome.

---

It is evident that the sub-population counterfactual outcome $Y(t|e) = Y_{S(e)}(t)$ is a composition of unit potential outcome variable $Y_i(t)$ and unit selection variable $S(e)$, with parameters computed by the network. Henceforth, for a unit $i$ with observed trace $X_i = x_i, T_i = t_i, Y_i = y_i$, the population-level counterfactual outcome $Y(t|e)$ and the individual-level counterfactual outcome $Y_i(t|e)$ are usually not identical. They are equal only when evidence $e$ is sufficient to identify this particularly unit, which constitutes an overly strong constraint.

The aforementioned algorithm implies that our DiscoModel, being a specialized neural network, is able to directly compute the counterfactual outcome $Y(t|e)$. This naturally prompts the question: Why does it specifically have to be a DiscoSCM? Is it feasible to design a DiscoModel under the SCM framework to conduct Layer 3 Valuations? Indeed, current mainstream causal modeling frameworks, both SCM and the PO framework, encounter challenges in this aspect. They grapple with the issue of degeneration stemming from the consistency assumption. This assumption imposes a stringent constraint that $Y(t)$ must consistent with $y$ when $e = [x, t, y]$ is observed. Such a constraint necessitates that the normal posterior $P(\cdot|e)$ for causal representation variable, with a non-degenerate normal distribution as the input of ActionNet, would yield a constant output. Integrating this requirement into the training of ActionNet results in degeneration, potentially culminating in an ActionNet where only the bias terms of the input layer are non-zero. From this viewpoint, given the network structure of DiscoModel, DiscoSCM becomes indispensable for those aspiring to learn a non-trivial ActionNet, as it extends the consistency assumption to a distribution-consistency assumption, thereby eliminating constraints that lead to the degeneration of ActionNet.

The primary objective in developing DiscoModel, which has been successfully achieved, was to predict the counterfactual outcome $Y(t|e)$, categorized as a Layer 3 valuation. This naturally leads to the computability of lower Layer Valuations. To substantiate this assertion mathematically, the expectation reduction method in Algorithm 2 serves as an example, from

which it can be deduced that the naive estimation for the counterfactual outcome could be:

$$\widehat{Y(t|e)} = \boldsymbol{\mu}_o(\boldsymbol{\mu}_r(e), t) \tag{8}$$

The understanding of the above formula specifically involves the following two derivations:

$$E[Y(t|e)] = E[Y(t)|e] = E[E[Y(t)|z_{S(e)}, e]] = E[\boldsymbol{\mu}_o(z_{S(e)}, t)]$$

and

$$E[z_{S(e)}] = \boldsymbol{\mu}_r(e)$$

Hence, the outcome variable given any evidence $e$ and intervention $do(t)$ can be estimated with DiscoModel. Consequently, setting appropriate $e$ and $t$ in the above estimation formula allows for the estimation of quantities Layer 1 $E[Y|X = \tilde{x}, T = \tilde{t}]$, Layer 2 $E[Y(t)|X = \tilde{x}]$, and Layer 3 $E[Y(t)|X = \tilde{x}, T = \tilde{t}, Y = \tilde{y}]$ as follows

$$\begin{cases} \widehat{E}[Y|\tilde{x}, \tilde{t}] & = \boldsymbol{\mu}_o(\boldsymbol{\mu}_r(\tilde{x}, \tilde{t}), \tilde{t}) \\ \widehat{E}[Y(t)|\tilde{x}] & = \boldsymbol{\mu}_o(\boldsymbol{\mu}_r(\tilde{x}), t) \\ \widehat{E}[Y(t)|\tilde{x}, \tilde{t}, \tilde{y}] & = \boldsymbol{\mu}_o(\boldsymbol{\mu}_r(\tilde{x}, \tilde{t}, \tilde{y}), t) \end{cases} \tag{9}$$

where $t$ can be a given treatment that differs from the observed trace. These estimations are referred to as the *naive predictions* of the counterfactual outcome.

To illustrate the capabilities of DiscoModel, consider a final example involving a user on an internet platform. This user, having received a high subsidy, exhibited high engagement. The question arises: Would the user still have demonstrated high engagement without the high subsidy, assuming all other conditions remained equal? Traditional causal frameworks would predict a 100% probability of high user engagement, which can be derived from the probability of consistency equals 1. However, DiscoSCM is capable of modeling non-trivial consistency probabilities, and allowing for heterogeneous consistency probabilities. In other words, DiscoModel has the ability to predict low user engagement, either by naive estimation $\boldsymbol{\mu}_o(\boldsymbol{\mu}_r(x, t, y), t)$ or other estimation methods. This capability can help us determine whether the high engagement of each user is brought about by high subsidies, providing invaluable insights for the industrial sector regarding individual incentivization strategies.

## 4 Simulation Study

This section starts with choosing suitable loss functions. Following that, we validate the model's proficiency in Layer Valuations using synthetic data. The final step involves assessing its effectiveness on a real-world dataset. For the implementation details, please refer to the following link: https://anonymous.4open.science/r/DiscoModel.

### 4.1 Loss Function

The basic idea is that the output prediction of DiscoModel given observation $e$ should be consistent with the observed outcome $y$. Hence, we might opt to minimize the following loss function:

$$\mathcal{L} = \sum_i (y_i - \widehat{Y(t_i|e_i)})^2 P(i) \tag{10}$$

where $e_i = [x_i, t_i, y_i]$ represents the observed trace of individual $i$. If we use naive predictions for $E[Y(t|e)]$ with equation 9 as an estimation for $Y(t|e)$, the loss function transforms into

$$\mathcal{L} = \sum_i (y_i - \boldsymbol{\mu}_o(\boldsymbol{\mu}_r(e_i), t_i))^2 P(i)$$

Another method is using the direct sampling strategy to obtain $\widehat{z_{S(e)}}$ and then estimate $Y(t|e)$. In our simulation, the former is used for prediction and evaluation, while the latter is employed for training the model.

## 4.2 Synthetic Data

In the data generation process (See Fig. 3), a causal representation $z$ is sampled from a bivariate normal distribution. Features $x$ are generated using $z$, an invertible matrix $A$, and noise. The treatment variable $t$ is uniformly sampled from $\{0, 1, 2\}$. The outcome $Y$ is derived from functions of $z$ and additional noise.

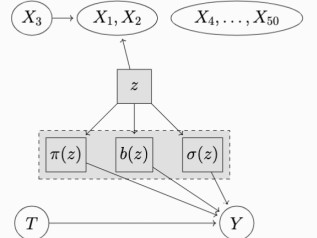

$z \sim \mathcal{N}(0, I)$

$X_3 \sim \mathcal{N}(0, 1), \quad X_4, \ldots, X_{50} \sim \text{Bernoulli}(p)$

$(X_1, X_2) = Az + \text{softplus}(X_3)\epsilon, \quad \epsilon \sim \mathcal{N}(0, 1)$

$T \sim \text{Uniform}\{0, 1, 2\}$

$\tau(z) = (z_1 + z_2 + 1)^+, \quad b(z) = z_1^2 + z_2^2, \quad \sigma(z) = |z_1 - z_2|$

$Y = \tau(z)T + b(z) + \sigma(z)\epsilon_y, \quad \epsilon_y \sim \mathcal{N}(0, 1)$

Figure 3: The equations (left) and graph (right) of the data generation process.

We trained a DiscoModel, and the Layer Valuations are exhibited in Fig. 4. This experiment validated that Layer Valuations can be effectively conducted with DiscoModel. Interestingly, it is observed that as more information is input, the precision of the outcome prediction increases, indirectly verifying the reasonableness of these calculations. Furthermore, our model demonstrates competitive performance in predicting ITE on real-world datasets, as substantiated through the Uplift Curve (See Fig. 7) and the evaluation metric AUUC, with details presented in Appendix A.4.

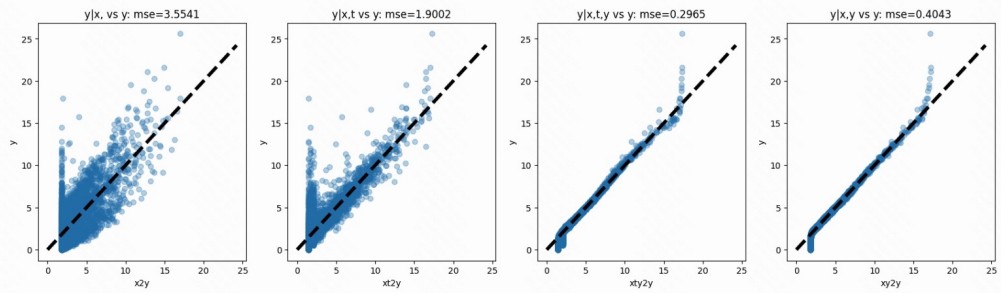

Figure 4: The naive estimation for counterfactual outcome under different scenarios: 1) $e = [x]$, 2) $e = [x, t]$ and $do(t)$, 3) $e = [x, t, y]$ and $do(t)$, 4) $e = [x, y]$.

## 4.3 Real-world RCT Dataset

Offering incentives to users, under cost constraints, is a common strategy employed by online platforms to enhance user engagement and increase platform revenue (Zhao & Harinen, 2019; Goldenberg et al., 2020; Ai et al., 2022). Personalized subsidies strategy require a model to to predict the ITE. We utilize a real-world dataset of 10,000 RCT samples on a certain internet platform which offers incentives to enhance user retention.

It can be observed that our DiscoModel exhibits comparable performance to the Causal-Forest algorithm (Athey & Wager, 2019) benchmark, assessed using the offline evaluation metric AUUC (See Appendix A.4) and Uplift Curve Fig. 5. Moreover, DiscoModel possesses potential advantages—it can directly address counterfactual questions, such as reflecting on whether high retention is a result of high subsidies. Other advantages include the ability to handle highly complex treatment forms, such as vectorized subsidies.

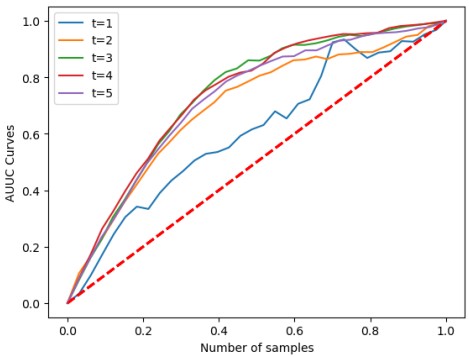

(a) Uplift curve for DiscoModel on real business data

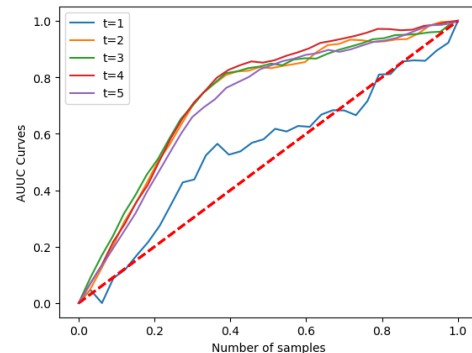

(b) Uplift curve for CausalForest on real business data

Figure 5: The evaluation results for DiscoModel and CausalForest on real business data

## 5 CONCLUSION

In DiscoModel, the fundamental design philosophy is to treat causality as invariance across heterogeneous units. The invariant causal relationship is encapsulated by ActionNet, which serves as a network to compute parameters for the outcome variable, taking heterogeneous unit representation as input. Consider a scenario where a user on a specific internet platform is observed with high subsidy and high retention. This raises the question, "if time were turned back and the same high subsidy offered, would a high retention outcome be guaranteed?" Your choice in this matter fundamentally determines whether you should employ DiscoSCM. Traditional causal modeling frameworks rely on the consistency rule, which would assure the same high retention in this situation. In contrast, DiscoSCM allows for predicting low retention with heterogeneous probabilities, aligning with the real-world scenario where decisions should be made assuming that one cannot control luck. Once a commitment is made to utilizing DiscoModel, which can be learned from the data, it thereby achieves practical and reasonable Layer 1/2/3 Valuations and allows for heterogeneous counterfactuals estimation across units—a feat that, to the best of our knowledge, has not been accomplished by any existing work.

### 5.1 DISCUSSION AND LIMITATIONS

Within the field of causality, there have been observed criticisms or a lack of acknowledgment regarding the practical value of Layer 3 Valuations, as evidenced by comments from Imbens (Imbens, 2020). One possible reason is the indeterminable nature of counterfactuals when lacking complete knowledge on structural equations, suggesting that they cannot be learned from observed or RCT data, which significantly limits their practicality. Our work is predicated on the acceptance of the value of Layer Valuations and the abandonment of the fundamental assumption of causal inference, the consistency assumption. This constitutes a notable limitation of this study, as it may encounter resistance from some researchers specializing in causal inference.

Advocates of consistency might contend: "An empirically minded scientist might say that, once we have data, the fact that some potential outcomes must equal their observed values in the data is a good thing; it is the information we have gained from the data." However, we prefer distribution-consistency and would argue: "An empirically minded scientist might prefer to maximize the likelihood of the observed values of variables in the data, rather than imposing equality constraints on them. Therefore, the observed value of potential outcomes in the data should be consistent with their distribution, rather than being strictly equated".

In conventional causal frameworks, the underlying basis of consistency assumption is essentially a form of Laplacian determinism, which is challenged by Dawid for causal modeling (Geneletti & Dawid, 2011; Dawid & Senn, 2023). It is plausible to conjecture that Pearl

might favor consistency over distribution-consistency given his advocacy for determinism (Pearl, 2009). If one firmly believes in determinism and denies the distribution-consistency assumption, a question arises: does such adherence lead to more convenient and practical models? After all, all models are wrong, but some are useful. Hence, our preference is to explore how a model, potentially incorrect, can be useful. While the philosophical correctness of determinism remains a subject of debate, we posit that deterministic modeling is unrealistic. A model serves as an abstraction of natural phenomena, and there exists irreducible computational complexity that cannot be compressed. DiscoSCM might function as a "reducible pocket"for this irreducible reality, serving as a probabilistic causal modeling approach for deterministic causal ground-truth.

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

## A  APPENDIX

### A.1  CAUSAL FRAMEWORKS BASED ON CONSISTENCY

There are two main frameworks for causal models: Potential Outcome (PO) and Structural Causal Models (SCM). Both of them are equivalent frameworks based on the consistency assumption (Pearl, 2011), but they can be challenging when computing counterfactuals, even for the simplest cases such as the correlation between potential outcomes with and without aspirin. The former can be interpreted as experimental causality, while the latter emphasizes causal mechanism modeling and derives the three Layers of information: associational, interventional and counterfactual.

The Potential Outcome (PO) framework, also known as the Rubin Causal Model (Holland, 1986), begins with a population of units. There is a treatment/cause that can take on different values for each unit. Each unit in the population is characterized by a set of potential outcomes $Y(t)$, one for each level of the treatment. In the simplest case with a binary treatment there are two potential outcomes, $Y(0)$ and $Y(1)$, but in other cases there can be more. Only one of these potential outcomes can be observed, namely the one corresponding to the treatment received:

$$Y^{obs} = \sum_t Y(t)\mathbf{1}_{T=t}. \tag{11}$$

This equation is derived from the consistency assumption that states

$$T^{obs} = t \Rightarrow Y(t) = Y^{obs}$$

The causal effects correspond to comparisons of the potential outcomes, of which at most one can be observed, with all the others missing. Paul Holland refers to this as the "fundamental problem of causal inference" (Holland, 1986).

We refer to and use the mathematical symbols in Xia et al. (2021) to provide a brief introduction to SCM. The basic semantic framework of our analysis rests on *structural causal models* (SCMs) , which are defined below.

**Definition 1** (**Structural Causal Model (SCM)**). *An SCM $\mathcal{M}$ is a 4-tuple $\langle \mathbf{U}, \mathbf{V}, \mathcal{F}, P(\mathbf{U}) \rangle$, where $\mathbf{U}$ is a set of exogenous variables (or "latents") that are determined by factors outside the model; $\mathbf{V}$ is a set $\{V_1, V_2, \ldots, V_n\}$ of (endogenous) variables of interest that are determined by other variables in the model – that is, in $\mathbf{U} \cup \mathbf{V}$; $\mathcal{F}$ is a set of functions $\{f_{V_1}, f_{V_2}, \ldots, f_{V_n}\}$ such that each $f_i$ is a mapping from (the respective domains of) $\mathbf{U}_{V_i} \cup \mathbf{Pa}_{V_i}$ to $V_i$, where $\mathbf{U}_{V_i} \subseteq \mathbf{U}$, $\mathbf{Pa}_{V_i} \subseteq \mathbf{V} \setminus V_i$, and the entire set $\mathcal{F}$ forms a mapping from $\mathbf{U}$ to $\mathbf{V}$. That is, for $i = 1, \ldots, n$, each $f_i \in \mathcal{F}$ is such that $v_i \leftarrow f_{V_i}(\mathbf{pa}_{V_i}, \mathbf{u}_{V_i})$; and $P(\mathbf{u})$ is a probability function defined over the domain of $\mathbf{U}$.*

Interventions and counterfactuals are defined through a mathematical operator called $do(\mathbf{x})$, which modifies the set of structural equations $\mathcal{F}$ to $\mathcal{F}_{\mathbf{x}} := \{f_{V_i} : V_i \in \mathbf{V} \setminus \mathbf{X}\} \cup \{f_X \leftarrow x : X \in \mathbf{X}\}$ while maintaining all other elements constant. Here we explain how an SCM $\mathcal{M}$ assigns values to the three layers in Pearl Causal Hierarchy in the following:

**Definition 2** (**Layers 1, 2 Valuations**). *An SCM $\mathcal{M}$ induces layer $L_2(\mathcal{M})$, a set of distributions over $\mathbf{V}$, one for each intervention $\mathbf{x}$. For each $\mathbf{Y} \subseteq \mathbf{V}$,*

$$P^{\mathcal{M}}(\mathbf{y}_{\mathbf{x}}) = \sum_{\{\mathbf{u} | \mathbf{Y}_{\mathbf{x}}(\mathbf{u}) = \mathbf{y}\}} P(\mathbf{u}), \tag{12}$$

*where $\mathbf{Y}_{\mathbf{x}}(\mathbf{u})$ is the solution for $\mathbf{Y}$ after evaluating $\mathcal{F}_{\mathbf{x}} := \{f_{V_i} : V_i \in \mathbf{V} \setminus \mathbf{X}\} \cup \{f_X \leftarrow x : X \in \mathbf{X}\}$.*
*The specific distribution $P(\mathbf{V})$, where $\mathbf{X}$ is empty, is defined as layer $L_1(\mathcal{M})$.*

**Definition 3** (**Layer 3 Valuation**). *An SCM $\mathcal{M} = \langle \mathbf{U}, \mathbf{V}, \mathcal{F}, P(\mathbf{U}) \rangle$ induces a family of joint distributions over counterfactual events $\mathbf{Y}_{\mathbf{x}}, \ldots, \mathbf{Z}_{\mathbf{w}}$, for any $\mathbf{Y}, \mathbf{Z}, \ldots, \mathbf{X}, \mathbf{W} \subseteq \mathbf{V}$:*

$$P^{\mathcal{M}}(\mathbf{y}_{\mathbf{x}}, \ldots, \mathbf{z}_{\mathbf{w}}) = \sum_{\substack{\{\mathbf{u} \mid \mathbf{Y}_{\mathbf{x}}(\mathbf{u}) = \mathbf{y}, \\ \ldots, \mathbf{Z}_{\mathbf{w}}(\mathbf{u}) = \mathbf{z}\}}} P(\mathbf{u}). \tag{13}$$

Each SCM $\mathcal{M}$ induces a causal diagram $G$ where every $V_i \in \mathbf{V}$ is a vertex, there is a directed arrow $(V_j \rightarrow V_i)$ for every $V_i \in \mathbf{V}$ and $V_j \in Pa(V_i)$. In the case of acyclic diagrams, which correspond to recursive SCMs, *do*-calculus (Pearl, 1995) can be employed to completely identify all Layer 2 expressions in the form of $P(y|do(x), c)$ (Huang & Valtorta, 2012). However, calculating counterfactuals at Layer 3 is generally far more challenging compared to Layers 1 and 2. This is because it essentially requires modeling the joint distribution of potential outcomes, such as the potential outcomes with and without aspirin. Unfortunately, we often lack access to the underlying causal mechanisms and only have observed traces of them. This limitation leads to the practical use of equation (equation 13) for computing counterfactuals being quite restricted.

## A.2 Probability of Causation

The probability of causation and its related parameters can be addressed by counterfactual logical (Pearl, 2009), three prominent concepts of which are formulated in the following :

**Definition 4** (**Probability of necessity (PN)**). *Let $X$ and $Y$ be two binary variables in a causal model $M$, let $x$ and $y$ stand for the propositions $X = true$ and $Y = true$, respectively, and $x'$ and $y'$ for their complements. The probability of necessity is defined as the expression:*

$$\begin{aligned} PN &\triangleq P(Y_{x'} = false | X = true, Y = true) \\ &\triangleq P(y'_{x'} | x, y) \end{aligned} \tag{14}$$

In other words, PN stands for the probability that event $y$ would not have occurred in the absence of event $x$, given that $x$ and $y$ did in fact occur. This counterfactual notion is used frequently in lawsuits, where legal responsibility is at the center of contention.

**Definition 5** (**Probability of sufficiency (PS)**).

$$PS \triangleq P(y_x | y', x') \tag{15}$$

**Definition 6** (**Probability of necessity and sufficiency (PNS)**).

$$PNS \triangleq P(y_x, y'_{x'}) \tag{16}$$

PNS stands for the probability that $y$ would respond to $x$ both ways, and therefore measures both the sufficiency and necessity of $x$ to produce $y$. Tian and Pearl (Tian & Pearl, 2000) provide tight bounds for PNS, PN, and PS without a causal diagram:

$$\max \left\{ \begin{array}{c} 0 \\ P(y_x) - P(y_{x'}) \\ P(y) - P(y_{x'}) \\ P(y_x) - P(y) \end{array} \right\} \leq \text{PNS} \tag{17}$$

$$\text{PNS} \leq \min \left\{ \begin{array}{c} P(y_x) \\ P(y'_{x'}) \\ P(x,y) + P(x',y') \\ P(y_x) - P(y_{x'}) + \\ + P(x,y') + P(x',y) \end{array} \right\} \tag{18}$$

$$\max \left\{ \begin{array}{c} 0 \\ \frac{P(y) - P(y_{x'})}{P(x,y)} \end{array} \right\} \leq \text{PN} \tag{19}$$

$$\text{PN} \leq \min \left\{ \begin{array}{c} 1 \\ \frac{P(y'_{x'}) - P(x',y')}{P(x,y)} \end{array} \right\} \tag{20}$$

In fact, when further structural information is available, we can obtain even tighter bounds for those parameters, as highlighted by recent research (Li & Pearl, 2019).

### A.3 DISCOSCMs: DISTRIBUTION-CONSISTENCY STRUCTURAL CAUSAL MODELS

In this section, we provide a brief introduction to distribution-consistency structural causal models (DiscoSCMs), and refer readers to Gong (2023) for more details.

On one hand, DiscoSCM can be viewed as an extension of PO framework, where the consistency assumption is replaced with the distribution-consistency assumption to leverage the advantages of PO in individual causality semantics. The *distribution-consistency* assumption can be expressed as follows:

**Assumption 1** (**Distribution Consistency**). *For any individual $i$ with treatment $T_i$, pre-treatment features $X_i$, and outcome $Y_i$:*

$$T_i = t \Rightarrow Y_i(t) \overset{d}{=} Y_i, \quad \forall i = 1, 2, \dots . \tag{21}$$

*where the symbol $\overset{d}{=}$ denotes identical distribution.*

This assumption implies that the randomness in the potential outcome $Y(t)$ given $T = t$ arises from both the selection of individuals and the exogenous uncertainty, as opposed to the PO framework which considers only unit selection. In fact, in the Distribution-consistency Structural Causal Model (DiscoSCM), the roles of units or individuals are pivotal. It makes a strict distinction between individual and population parameters, fundamentally considering individual-level valuations as primitives and population-level valuations as derivatives.

On the other hand, DiscoSCM can also be perceived as a generalization of Structural Causal Model (SCM), facilitating the representation of Layer Valuations.

**Definition 7** (**Distribution Consistency Structural Causal Model (DiscoSCM)**). *A DiscoSCM $\mathcal{M}$ is a 4-tuple $\langle \mathbf{U}, \mathbf{V}, \mathcal{F}, P(\mathbf{U}) \rangle$, where $\mathbf{U}$ is a set of exogenous variables (or "latents") that are determined by factors outside the model; $\mathbf{V}$ is a set $\{V_1, V_2, \dots, V_n\}$ of (endogenous) variables of interest that are determined by other variables in the model – that is, in $\mathbf{U} \cup \mathbf{V}$; $\mathcal{F}$ is a set of functions $\{f_{V_1}, f_{V_2}, \dots, f_{V_n}\}$ such that each $f_i$ is a mapping from (the respective domains of) $\mathbf{U}_{V_i} \cup \mathbf{Pa}_{V_i}$ to $V_i$, where $\mathbf{U}_{V_i} \subseteq \mathbf{U}$, $\mathbf{Pa}_{V_i} \subseteq \mathbf{V} \setminus V_i$, and the entire set $\mathcal{F}$ forms a mapping from $\mathbf{U}$ to $\mathbf{V}$. That is, for $i = 1, \dots, n$, each $f_i \in \mathcal{F}$ is such that $v_i \leftarrow f_{V_i}(\mathbf{pa}_{V_i}, \mathbf{u}_{V_i})$; and $P(\mathbf{u})$ is a probability function defined over the domain of $\mathbf{U}$. A mathematical operator called $do(\mathbf{x})$, which modifies the set of structural equations $\mathcal{F}$ to $\mathcal{F}_{\mathbf{x}} := \{f_{V_i} : V_i \in \mathbf{V} \setminus \mathbf{X}\} \cup \{f_X \leftarrow x : X \in \mathbf{X}\}$ while maintaining the same endogenous uncertainty as $\mathbf{U}$, induces a submodel $\langle \mathbf{U}(\mathbf{x}), \mathbf{V}, \mathcal{F}_{\mathbf{x}}, P(\mathbf{u}) \rangle$.*

It's important to note that the formulation distinction between DiscoSCM and SCM lies in the construction of the submodel induced by the *do*-operator, which changes $\mathbf{U}$ to $\mathbf{U}(x)$ that is referred as the potential noise with same distribution as $\mathbf{U}$. This modification enables the prediction of an average score for retaking a test for a specific individual who, with average ability, takes a test and achieves an exceptionally high score due to good luck. Similarly, interventions and counterfactuals are defined in the following manner:

**Definition 8 (Layer 1, 2, 3 Valuation).** *A DiscoSCM $\mathcal{M} = \langle \mathbf{U}, \mathbf{V}, \mathcal{F}, P(\mathbf{U}) \rangle$ induces a family of joint distributions over potential outcomes $\mathbf{Y}\mathbf{x}, \ldots, \mathbf{Z}_{\mathbf{w}}$, for any $\mathbf{Y}, \mathbf{Z}, \ldots, \mathbf{X}, \mathbf{W} \subseteq \mathbf{V}$:*

$$P^{\mathcal{M}}(\mathbf{y}) = \sum_{\{\mathbf{u} \mid \mathbf{Y}(\mathbf{u})=\mathbf{y}\}} P(\mathbf{u}), \tag{22}$$

$$P^{\mathcal{M}}(\mathbf{y_x}) = \sum_{\{\mathbf{u_x} \mid \mathbf{Y_x}(\mathbf{u_x})=\mathbf{y}\}} P(\mathbf{u_x}), \tag{23}$$

$$P^{\mathcal{M}}(\mathbf{y_x}, \ldots, \mathbf{z_w}) = \sum_{\substack{\{\mathbf{u_x} \ldots, \mathbf{u_w} \mid \mathbf{Y_x}(\mathbf{u_x})=\mathbf{y}, \\ \ldots, \mathbf{Z_w}(\mathbf{u_w})=\mathbf{z}\}}} P(\mathbf{u_x}, ..., \mathbf{u_w}). \tag{24}$$

By definition, it's evident that all Layer 1 and Layer 2 valuations within the SCM and DiscoSCM frameworks are equivalent. However, Layer 3 valuations exhibit differences. These differences are best exemplified by considering the counterfactual parameter, PNS, at individual-level. In the SCM framework, this parameter degenerates to either 0 or 1, whereas in the DiscoSCM framework, it can take any value between 0 and 1, for a specific individual $i$. In fact, the degenerative probability of causation parameters prevalent in the SCM framework no longer degenerate in DiscoSCM. This non-degeneration provides convenience for utilizing and modeling these parameters, thereby allowing for the definition of a novel parameter within DiscoSCM:

**Definition 9 (Probability of Consistency (PC)).**

$$PC \triangleq P(y_x|y, x) \tag{25}$$

It is evident that PC degenerates to constant 1 in the SCM framework and is thus a parameter that only holds significance within the DiscoSCM framework. Individual-level valuations are primitive, and population-level valuations are derivation. Here we present the following procedure for the population-level valuations:

**Theorem 1 (Population-Level Valuations).** *Consider a population where A represents a counterfactual event (such as being a complier), and c represents observed-variable conditions (e.g., observed $T = t$, $Y = y$). Then, the population-level valuations of the form $P(A|c)$ can be computed via the following three-step algorithm:*

**Step 1 (Abduction):** *From the context c, derive a individual selector S to define a population with a distribution $P'$. A sample selector can typically be defined by the posterior over the index-set of samples given the context c and a uniform prior.*

**Step 2 (Valuation):** *Compute $P(A_i)$ as Layer valuations in Def. 8 for each individual $i$.*

**Step 3 (Reduction):** *Obtain the population-level $P(A)$ by summing over all individuals, which can be expressed as follows:*

$$P(A|c) = \sum_i P(A_i)P'(i) \tag{26}$$

The valuation step involves the computation of individual-level counterfactuals, which is often infeasible due to the *Indeterminable Counterfactuals* problem. This issue describes a scenario where individual-level counterfactual information remains elusive, relying solely on data, even in the case of the most comprehensive and ideal dataset. To address this challenge, DiscoSCM incorporates an independent potential noise assumption.

**Definition 10.** *A DiscoSCM* $\mathcal{M} = \langle \mathbf{U}, \mathbf{V}, \mathcal{F}, P(\mathbf{U}) \rangle$ *with independent potential noises induces a family of joint distributions over potential outcomes* $\mathbf{Y_x}, \ldots, \mathbf{Z_w}$*, for any* $\mathbf{Y}, \mathbf{Z}, \ldots, \mathbf{X}, \mathbf{W} \subseteq \mathbf{V}$*, satisfying:*

$$P(\mathbf{u_x}, ..., \mathbf{u_w}) = P(\mathbf{u_x}) \cdots P(\mathbf{u_w}) \tag{27}$$

The term "independent potential noises" refers to the independence among the exogenous noises across different counterfactual worlds. Combined with Eq. equation 24, the following theorem for individual-level Layer 3 valuations can be derived:

**Theorem 2.** *For potential outcomes* $\mathbf{Y_x}, \ldots, \mathbf{Z_w}$ *in a DiscoSCM* $\mathcal{M} = \langle \mathbf{U}, \mathbf{V}, \mathcal{F}, P(\mathbf{U}) \rangle$ *with independent potential noises:*

$$P^{\mathcal{M}}(\mathbf{y_x}, \ldots, \mathbf{z_w}) = P^{\mathcal{M}}(\mathbf{y_x}) \cdots P^{\mathcal{M}}(\mathbf{z_w}) \tag{28}$$

This theorem elegantly reduces Layer 3 valuations to Layer 2 valuations, facilitating the identification of individual-level counterfactuals even in the absence of knowledge about structural equations, which are typically a prerequisite in ordinary SCMs. In fact, the correlation patterns of potential noises determine the type of DiscoSCM, as can be illustrated by the following example:

**Example 1.** *Consider a DiscoSCM for the outcome* $Y$ *with features* $X_0, X_1, X_2$ *and binary treatment* $T$*:*

$$Y = 0.5I[X_0 = 1] \cdot (T + 1) + 0.1X_2 \cdot \epsilon$$
$$Y(t) = 0.5I[X_0 = 1] \cdot (t + 1) + 0.1X_2 \cdot \epsilon(t)$$

*where* $\epsilon, \epsilon(t) \sim N(0, 1), t = 0, 1$ *denote the noise and potential noises respectively. Fig. 6 showcases an RCT dataset produced by this DiscoSCM.*

## A.4 Experiment Results

**Setting.** we assign the layer valuations loss weights as $[1, 1, 5, 5]$ to prioritize the model's performance in predictions absent of information in $y$, thereby mitigating overreliance on $y$ data. The causal representation $z$ dimensions are 25 for synthetic data and 30 for real business data. We employ a learning rate of 0.001 and a random seed of 1. During training and evaluation process, we refer to Mean Squared Error (MSE) loss. Notably, a patience of 200 for early stopping is applied to synthetic data 2 to avoid premature training termination. The experiment with synthetic data can be replicated using the provided code, while the experiment with real-world data can only be replicated after obtaining authorization to disclose the data. For the code related to the experiments, please refer to the following link: https://anonymous.4open.science/r/DiscoModel

**Results.** The figure below illustrates the evaluation results of DiscoModel and CausalForest on synthetic data and real RCT data, respectively. The AUUC scores of DiscoModel on synthetic data are $[0.6796, 0.7059]$, with different figures corresponding to different treatments. For comparison, the ground truth AUUC is $[0.7570, 0.7649]$, while CausalForest yields an AUUC score of $[0.6739, 0.6830]$, indicating that DiscoModel's performance is as good as or slightly better than that of CausalForest on synthetic data. When trained and evaluated on real business data, DiscoModel also demonstrates a similar capability to CausalForest, yielding an AUUC score of $[0.6046, 0.6938, 0.7356, 0.7386, 0.7199]$, compared to $[0.5380, 0.7234, 0.7298, 0.7392, 0.7098]$ generated by CausalForest.

The figure below presents the evaluation results of DiscoModel and CausalForest on both synthetic data and real RCT data. For synthetic data, DiscoModel achieves AUUC scores of $[0.6796, 0.7059]$, with different figures corresponding to distinct treatments. In comparison, the ground truth AUUC is $[0.7570, 0.7649]$, and CausalForest attains an AUUC score of $[0.6739, 0.6830]$. This suggests that the performance of DiscoModel is comparable to, if not slightly superior to, that of CausalForest on synthetic data. When applied to real business data, DiscoModel continues to exhibit a competitive capability, securing an AUUC score of $[0.6046, 0.6938, 0.7356, 0.7386, 0.7199]$, as opposed to $[0.5380, 0.7234, 0.7298, 0.7392, 0.7098]$ achieved by CausalForest.

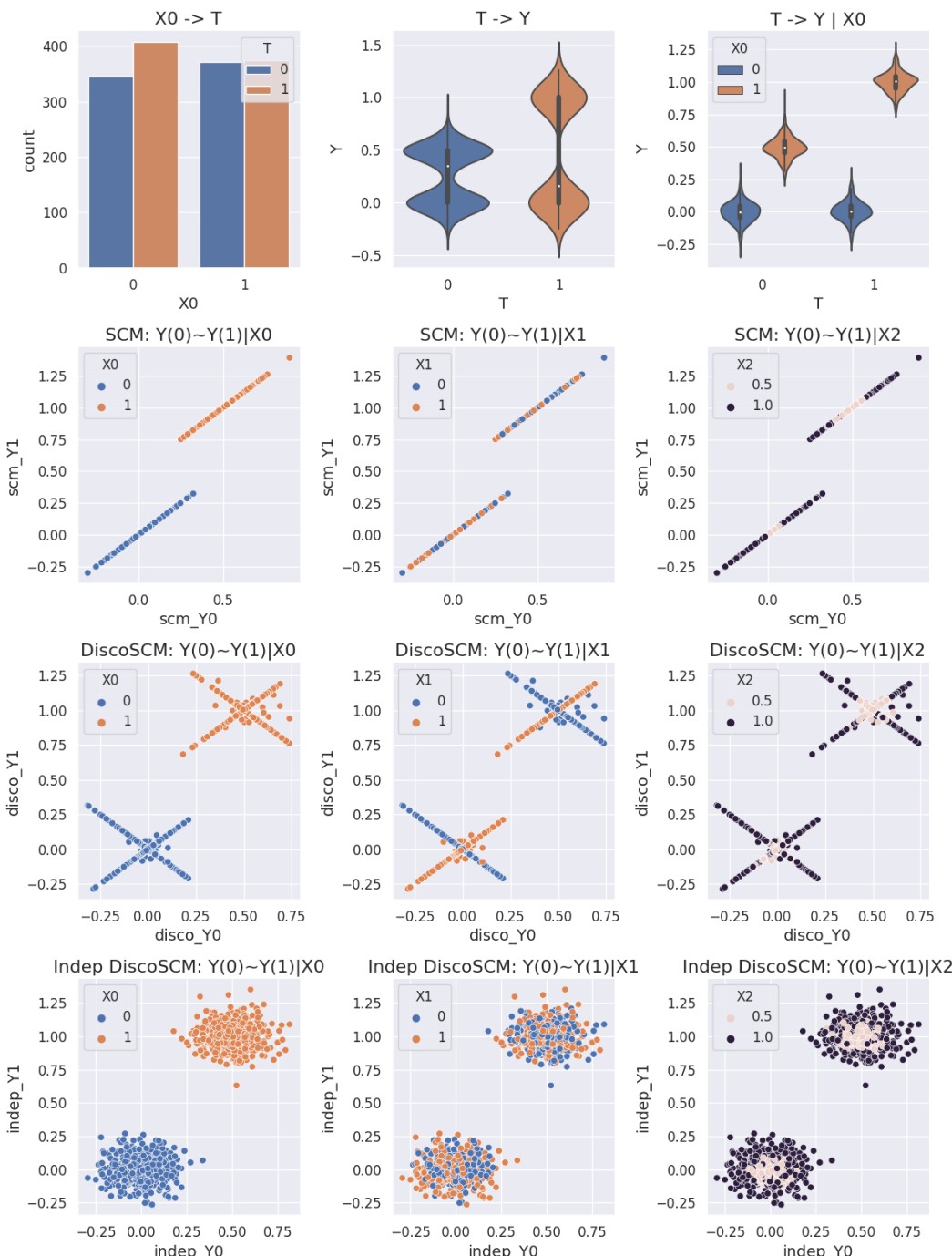

Figure 6: DiscoSCM with heterogeneous causal effects, potential noise correlations, and consistency probabilities. The type of this DiscoSCM depends on the correlation pattern among potential noises $\epsilon(t)$: if the correlation coefficient is 1, it is an ordinary SCM; if there is some correlation, it is a general DiscoSCM with indeterminable counterfactuals; if the potential noises are independent, it becomes a DiscoSCM where Layer 3 individual-level counterfactuals can be reduced to Layer 2 valuations. Specifically, for individual counterfactual parameters $corr(Y_i(0), Y_i(1))$, the second row of Fig. 6 shows that it is always 1 in the SCM, while the third row reveals that its value lies between 0 and 1 in the general DiscoSCM, showing heterogeneity according to $X_1$. The fourth row of Fig. 6 demonstrates that this parameter is always 0 in a DiscoSCM with independent potential noise. To summarize, when the correlation between potential noises is 1, as is the case in SCM, knowledge of all structural equations is required to solve for counterfactuals. When potential noises exhibit some correlation, neither randomized controlled trial (RCT) or observational data can help recover related counterfactual parameters. Conversely, when potential noises are independent, Layer 3 valuation can be reduced to Layer 2, allowing them to typically be identified from the data. See code in link.

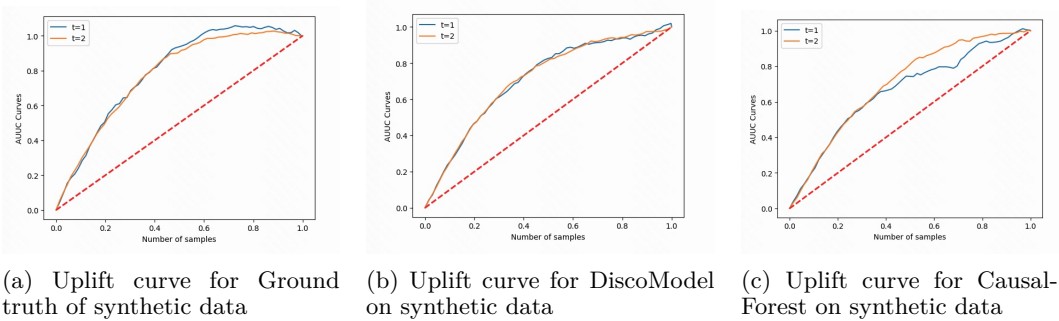

(a) Uplift curve for Ground truth of synthetic data

(b) Uplift curve for DiscoModel on synthetic data

(c) Uplift curve for Causal-Forest on synthetic data

Figure 7: The evaluation results for DiscoModel and CausalForest on synthetic data

