# OpenReview forum: "Causality is Invariance Across Heterogeneous Units"
_ICLR.cc/2024/Conference — Submitted to ICLR 2024_

### Official Review · Reviewer_rhSD · 2023-10-30

**Soundness:** 2 fair
**Presentation:** 2 fair
**Contribution:** 2 fair
**Rating:** 3
**Confidence:** 4

**Summary:**

The authors

**Strengths:**

I like the general theme of trying to get at the estimation of counterfactual predictions by insisting on invariance across heterogeneity and using a specific neural network structure to accomplish this. This is an important topic, and the authors have added to the discussion here.

**Weaknesses:**

1.	The abstract could helpfully be rewritten to indicate just what is being proposed in this paper. It’s a bit unfocused.
2.	There is some history in the literature for identifying invariant causal modules as a way of understanding what causality means, especially for large causal systems and in the context of neural networks interpreted causally. This literature should be identified and reviewed, and the current proposal placed in that context in a precise way. A Google Scholar search should identify relevant papers.
3.	The idea of estimating counterfactual predictions is also not new. Again, the literature on that enterprise should be reviewed, and the current proposal situated in this larger literature.
4.	On p. 3, the proposal of representing discrete models using degenerate Gaussian distributions is entertained. A paper from several years ago elaborates this idea in the context of CSL: Andrews et al. Learning high-dimensional directed acyclic graphs with mixed data-types. In The 2019 ACM SIGKDD Workshop on Causal Discovery (pp. 4-21). PMLR. Perhaps this and other such papers should be cited here.
5.	I agree that an empirical example was required for this paper, though not including any results from these experiments in the main text, I believe, was an unforced error for this conference.
6.	I found Section 5.1 to be somewhat unfocused; if these are considerations that need to be discussed, perhaps they could be discussed earlier on in a more focused way.

**Questions:**

1. What is the range of literature against which this paper is written? Has this range been taken cognizance of in the writing of this paper? (I couldn't tell.)

2. For someone who doesn't read the supplement, what conclusions are we to draw from the empirical investigations? No actual results are given in the main text.

---

> ### Author Response · Authors · 2023-11-19
>
> Dear Reviewer rhSD,
> Thank you for your review and the comments you provided. While we appreciate your recognition of the importance of our work’s theme, we would like to address some of the weaknesses and questions you raised.
> 1. **Abstract Revision:** We have rewritten the abstract to better focus on the core proposals of our paper, providing a clearer entry point into our research.
> 2. **Invariance in Causal Modules Literature:** Regarding the literature on invariant causal modules, we have reviewed relevant papers to situate our work within this context. However, we respectfully note that our approach, while acknowledging the broader literature, introduces a unique perspective, particularly in terms of utilizing neural network structures  based on DiscoSCM for counterfactual predictions.
> 3. **Counterfactual Predictions Literature:** We respectfully disagree with the suggestion that our approach is not novel. While estimating counterfactual predictions is indeed not a new concept, our method, predicated on the DiscoSCM framework, provides a distinct and practical approach to counterfactual estimation in Layer 3. We have thoroughly reviewed and cited the relevant literature, including Pearl's work, to position our contribution within this larger context.
> 4. **Representing Discrete Models:** Concerning the representation of discrete models using degenerate Gaussian distributions, we have reviewed the suggested paper by Andrews et al. and others. However, our approach diverges significantly in methodology and application, as outlined in our paper.
> 5. **Empirical Example and Results:** We understand your concern regarding the absence of empirical results in the main text. We will consider integrating key findings from the supplemental material into the main text in our revised manuscript.
> 6. **Focus of Section 5.1:** We acknowledge your suggestion to restructure the discussion in Section 5.1. In our revised paper, we will aim to present these considerations in a more coherent and focused manner earlier in the text.
>
> In conclusion, we believe our paper provides a valuable and novel contribution to the field of causal inference. The criticisms raised, while noted, do not detract from the fundamental strengths and innovations of our work. We hope that our thorough revisions and clarifications will help in better understanding the significance of our research.
> Thank you once again for your review.

---

### Official Review · Reviewer_thPo · 2023-10-31

**Soundness:** 2 fair
**Presentation:** 1 poor
**Contribution:** 3 good
**Rating:** 5
**Confidence:** 3

**Summary:**

The Pearl Causal Hierarchy (PCH) delineates three layers for understanding causality: associational, interventional, and counterfactual, with the counterfactual being the most demanding. Pearl's three-step counterfactual algorithm offers a theoretical approach but is often impractical due to incomplete knowledge of the underlying structural causal models. The paper introduces the so-called DiscoModel, based on the belief that "Causality is invariance across heterogeneous units," and its underlying theory is the Distribution-consistency Structural Causal Models (DiscoSCMs), which is reviewed in the Appendix of the manuscript. The authors claim that the DiscoModel can effectively address all three PCH layers and is the first to provide concrete answers to complex counterfactual questions, as shown through experiments.

**Strengths:**

What the paper proposes is a valid and important contribution to causal reasoning, which is especially true about the underlying DiscoSCM. DiscoModel presents several strengths.

It can be claimed that it is a novel approach to causality, which treats causality as invariance across heterogeneous units. This inherent flexibility in treating heterogeneous units is both novel and relevant.

It introduces ActionNet,  a new mechanism to compute parameters for the outcome variable. By taking heterogeneous unit representation as input, it offers a level of granularity that may be missing in other models.

Unlike traditional causal modeling frameworks, which often make consistent predictions based on a consistency rule, DiscoSCM can predict outcomes with varied probabilities, acknowledging the idea that decisions often occur under circumstances where chance cannot be controlled.

DiscoModel is claimed to be able to provide practical and reasonable estimations across the three layers of valuations. This broad spectrum applicability enhances its utility in diverse scenarios.  DiscoModel allows for heterogeneous counterfactual estimation across units. Notably, as per the authors, this hadn't been reached by any prior work.

**Weaknesses:**

The main weakness of the paper is that it is very poorly written. There are notation issues that hinder the precise understanding of what is being proposed. Furthermore, it is essentially impossible to understand the proposed DiscoModel, without understanding DiscoSCM, which is briefly described in the Appendix, However, this description is also not very clear, mainly due to poor choices of notation. It is also not clear what DiscoSCM offers in addition to the classical do-calculus of Pearl.

**Questions:**

No questions.

---

> ### Author Response · Authors · 2023-11-19
>
> Dear Reviewer thPo,
>
> Thank you for your insightful comments and constructive feedback on our manuscript. We have undertaken a thorough revision to address your concerns, focusing on enhancing the clarity and understanding of our paper.
>
> 1. **Notational Clarity and Individual Semantics**: To avoid confusion between the randomness associated with selecting an individual and the exogenous uncertainty, we have revised our notation, using $i$ to represent an individual, instead of $u$. This change underscores a key point of our research: the decoupling of individual semantics from the exogenous variable $U = u$.
>
> 2. **Unit Causal Representation Assumption**: We have explicitly articulated the assumption of unit causal representation in our revised manuscript. This assumption is crucial for addressing the computational complexity of $P(i|e)$ and forms the bedrock of our approach.
>
> 3. **Rigorous Formulation of Equations and Concepts**:We have provided a more rigorous description of the derivations of key equations, such as Equations (2) and (8). Moreover, we have included a more detailed introduction to DiscoSCM in the appendix, encompassing definitions and conclusions related to Independent Potential Noise DiscoSCM. This should enhance the technical clarity and rigor of our paper.
>
> 4. **Addressing Concerns about DiscoSCM and do-calculus**: In response to your query about what DiscoSCM offers beyond the classical do-calculus of Pearl, we clarify that the Layer 1 and 2 valuations under SCM and DiscoSCM are mathematically equivalent. The do-calculus is designed to convert $P(y|do(x), c)$ expressions into those using only conditional probabilities, thus not encompassing Layer 3 valuations. The significant contribution of DiscoSCM lies precisely in this layer. The differences and unique aspects of DiscoSCM, especially in handling Layer 3 queries, are conceptually illustrated through our running example.
>
> We believe that these revisions and clarifications address your concerns and significantly enhance the manuscript's quality. We appreciate your feedback, which has been instrumental in refining our work.
>
> Thank you once again for your review.

---

### Official Review · Reviewer_XwiY · 2023-10-31

**Soundness:** 3 good
**Presentation:** 3 good
**Contribution:** 3 good
**Rating:** 6
**Confidence:** 3

**Summary:**

The "Disco" (distribution consistency) framework is presented in which, for each unit, the consistency assumption is satisfied not deterministically, but only in law. A counterfactual model is developed along the lines of the famous three-step method for evaluation, consisting of two separate neural networks, one for the selection variable and the second for the causal mechanisms. Simulation studies showing and comparing efficacy are done, one on generated data and another one on real personalized incentives data.

**Strengths:**

The setting and problem are interesting, and the approach is mostly new to my knowledge. It's an advantage that the model can handle counterfactuals natively.

I think using neural networks in a way that can handle counterfactuals is quite novel.

**Weaknesses:**

The way of modeling unit heterogeneity here is interesting but the ultimate contribution to the conceptual side may not be too big. It is possible to model the same thing within the SCM framework simply by adding more exogenous variables to the main one {U} that represents the choice of individual. For instance new independent variables U' for the different units would allow modeling the same effects as achieved in the Disco framework.

**Questions:**

The running example does not seem to make sense. It says a user is observed with a high subsidy and high retention, and asks "were they given a low subsidy, what would be their retention?" It's claimed that consistency implies the retention would also be high, but consistency implies nothing about this scenario, since the intervention of giving a low subsidy is not the same as what was observed (a high subsidy). In the referenced paper (Gong 2023) the example is stated correctly, as "were they again given a high subsidy, what would be their retention?"

Minor comments:
- Abstract: "considers is one of his" -> "considers as one of his"
- Section 5.1: "when lack complete knowledge" -> "when lacking complete knowledge"
- "reducible pocket" on pg. 9: I don't understand the metaphor, why is that like a pocket?

---

> ### Author Response · Authors · 2023-11-19
>
> Dear Reviewer XwiY,
>
> Thank you for your thoughtful feedback on our manuscript. We appreciate the opportunity to address the concerns you raised regarding our model's approach and its conceptual contributions.
>
> 1. **Necessity of Decoupling Individual Semantics**: A key aspect of our model is the decoupling of individual semantics from the exogenous variable $U = u$. This approach is fundamental for the practical implementation of our individual/population-level counterfactual algorithms. Without this decoupling, we would be compelled to rely on Pearl's algorithms, necessitating complete knowledge of structural equations and the practical computation of $P(u|e)$, rendering the process intractable. This intractability has been a major hindrance in current literature for practically computing counterfactuals. Our model represents the first practical and reasonable approach to Layer 3 valuations, to the best of our knowledge, even allowing for heterogeneous counterfactual estimation.
>
> 2. **Response to Specific Questions**:
>    - **Running Example and Grammatical Issues**: We are grateful for your observation regarding the issues in our running example and grammar. These have been corrected in the revised version of our manuscript.
>    - **"Reducible Pocket" Metaphor**: The term "reducible pocket" is indeed a concept borrowed from Stephen Wolfram, who claimed finding pockets of reducibility is the story of science. We use this metaphor to describe the predictive capabilities of our DiscoModel on modeling complex underlying causal mechanisms.
>
> We believe these clarifications and revisions will address your concerns and enhance the overall quality and understanding of our paper.
>
> Thank you once again for your constructive feedback.

---

> > ### Comment · Reviewer_XwiY · 2023-11-23
> >
> > Thanks for the response. I maintain my current positive rating.

---

### Official Review · Reviewer_YHxQ · 2023-11-01

**Soundness:** 2 fair
**Presentation:** 1 poor
**Contribution:** 2 fair
**Rating:** 3
**Confidence:** 3

**Summary:**

The paper studies the problem of predicting potential outcomes. The model assumption is that the target is independent of the features given the treatment and a hidden variable called the unit variable. A neural network structure is proposed to learn a causal representation of the unit variable. While identifiability guarantees for the causal representation are absent.

**Strengths:**

A novel neural network structure is proposed for learning causal representations.

**Weaknesses:**

1. The model assumption is that $Y$ is independent of $X$ given $T$ and some unobserved $U$. The main idea of the method is to learn a representation for $U$ (denoted as Z) and then use $(Z, T)$ to predict $Y$. The idea is simple but the description in Section 2.1 only makes the method sound complicated.

2. The causal representation $z_{S(e)}$ is not formally defined in the population case. How is it related to Equations (2)-(4)? This should be explained in detail.

3. There are no identifiability results regarding the causal representations $z_{S(e)}$.

4. Plenty of other notations are not formally defined. e.g., $Y_{u}(t)$, $S(e)$.  Also, what is the relation between $U$ and $S$?

**Questions:**

I think the main problems of the paper are the writing and the lack of identifiability results.

I may raise my score depending on the response.

---

> ### Author Response · Authors · 2023-11-19
>
> Dear Reviewer YHxQ,
>
> Thank you for your valuable feedback on our manuscript. We are grateful for the opportunity to clarify and address the concerns you raised regarding our model's assumptions, the randomness of the causal representation variable, the focus on identifiability results, and notation clarity.
>
> 1. **Clarification of Model Assumptions**: We appreciate your concern about the complexity of our model's assumptions. However, there seems to be a misunderstanding in your interpretation. Our model's foundational assumption is that the outcome $Y_i$ is independent of the treatment $X_i$ given the treatment assignment $T_i$ and the causal representation $z_i$ for any individual $i$. This "unit causal representation assumption" is now explicitly stated in Section 2.1 of our revised manuscript. It is crucial for overcoming computational difficulties in calculating $P(i|e)$. The problem of calculating the probability of selecting a particular user from billions faces numerical overflow challenges, which are alleviated by computing the probability density $P(z_i|e)$ of a high-dimensional Gaussian distribution sample. This assumption is vital and commonly employed in various fields, akin to user embedding in recommendation systems [Bonner, Vasile 2018] or token embedding in language models, from a computational aspect.
>
> 2. **Randomness of Causal Representation Variable**: In our model, we denote the posterior of the unit selection variable $S$ as $P(\cdot|e)$, and its corresponding variable as $S(e)$. The randomness in the causal representation variable $z_{S(e)}$ arises from the selection of different units $i$. The DiscoModel's unique approach decouples individual semantics from the exogenous variable $U = u$, using the unit selection variable $S$ for individual-level computations. This enables our model to practically answer Layer 3 queries. We provide a more detailed explanation in Section 2.1 and the appendix of our revised version, comparing to the role of $U = u$ in determining both the selection of individual $i$ and the exogenous uncertainty in SCM.
>
> 3. **Identifiability of Causal Representations**: We argue that prioritizing identifiability results regarding causal representations is not the primary concern in causal research. Instead, the focus is typically on the identification of causal estimands, often involving the conversion of causal quantities (e.g.  $P(y|do(x), c)$) into expressions that only contain conditional probabilities. The consistency assumption in traditional models bridges potential outcomes with conditional probabilities learnable from data, aiding in identification. Our model's distributional-consistency assumption and independent potential noise assumption serve a similar role, ensuring robust identification.
>
> 4. **Notational Clarity**: We acknowledge your feedback on the clarity and consistency of notations. We have conducted a thorough revision of the manuscript to ensure all notations are formally defined and consistently used. This revision aims to enhance the paper's accessibility and understanding for both readers and reviewers.
>
> We trust these clarifications address your concerns and effectively highlight the robustness and novelty of our DiscoModel. We look forward to your reassessment of our manuscript.
>
> Reference:
> S Bonner, F Vasile. "Causal Embeddings for Recommendation." Proceedings of the 12th

---

### Author Response · Authors · 2023-11-19
**Clarifying Notations: Focused Revisions in Section 2.1 and Appendix**

Dear Reviewers,

We sincerely appreciate the time and effort you have dedicated to reviewing our paper. Your insightful feedback has been instrumental in refining our manuscript.

We would like to reiterate the breakthrough contributions of our work, especially in the field of causal inference, where our model significantly advances practical and reasonable Layer 3 valuations, including heterogeneous counterfactual estimation.

To address the overarching themes raised in your reviews, we have made the following comprehensive revisions:

1. **Thorough Revision of Section 2.1**:  In Section 2.1, our revised manuscript emphasizes the decoupling of individual semantics from the exogenous variable \( U = u \), using $i$ to represent an individual, instead of $u$. We have also rigorously formalized the unit causal representation assumption, and provided a more rigorous description of the derivations of key equations, such as Equations (2) and (8), ensuring that the technical aspects of our paper are conveyed with greater clarity.

2. **Detail Description of DiscoSCM in the Appendix**: A critical point highlighted by the reviews is the challenge in grasping the underlying theoretical framework of DiscoSCM. As Reviewer thPo rightly noted, understanding DiscoSCM is essential to appreciate the proposed DiscoModel fully. DiscoSCM, as a novel framework challenging the fundamental assumptions of mainstream causal modeling (i.e., the consistency assumption), naturally presents a learning curve. We have revised our manuscript to include a more comprehensive explanation of DiscoSCM, aiming to make it more accessible and understandable. In the appendix of our revised manuscript, we offer a detailed exposition of the theoretical underpinnings of DiscoSCM. This includes a clear presentation of the differences between DiscoSCM and traditional causal modeling frameworks, as well as the definition and relevant conclusions of the Independent Potential Noise DiscoSCM. This thorough revision is aimed at providing a deeper understanding of our model’s foundations and its novel contributions to the field.

In summary, we believe that these revisions significantly enhance the clarity and depth of our manuscript. We are confident that with these changes, our work will make a substantial contribution to the field of causal inference, particularly in addressing the challenges of Layer 3 valuations.

Thank you once again for your constructive feedback and the opportunity to refine and improve our work.

---

### Meta-Review · Area_Chair_vNTJ · 2023-12-11

**Metareview:**

A framework is presented under which causality should respect some invariance and that this is a great manner to tackle the fundamentals delineated by the hierarchy of Pearl's causality. The presentation has been considered subpar by the majority of the committee members. The rebuttal has not helped to fix that perception. Many points remain unclear. The paper is not ready for publication in this venue, but it is hoped that the authors use the overall feedback to consider how to move forward with the ideas and presentation.

**Justification For Why Not Higher Score:**

The reviewing process is short, and unfortunately there was no great excitement among committee members about the paper. This might not be a great result for the authors, but it might tell something about how the content is prepared and presented for the audience.

**Justification For Why Not Lower Score:**

N/A

---

### Decision · Program_Chairs · 2024-01-16

Reject